# Understanding the Support Needs of Family Caregivers Living with Severe Developmental Disability: An Interpretive Phenomenological Analysis

**DOI:** 10.3390/healthcare13202550

**Published:** 2025-10-10

**Authors:** Anna McStravick, Rosanna Cousins

**Affiliations:** 1Health and Human Services, Trellis Australia, Melbourne 3205, Australia; annamcstravick99@gmail.com; 2School of Psychology, Liverpool Hope University, Liverpool L16 9JD, UK

**Keywords:** associative disability, impairment, family carer, quality of life, stigma, Northern Ireland

## Abstract

**Background/Objectives:** Living with a disabled family member has extensive implications for the whole family involved in their care, and there is dependency on healthcare support for maintaining quality of life. This qualitative study, conducted in Northern Ireland, investigated the support needs of different family members living with a severely impaired individual across the lifespan. A key objective was to identify support needs for intervention. **Methods:** In-depth semi-structured interviews were conducted to obtain data from eight mothers, fathers, sisters and brothers of a profoundly disabled child or sibling. Data was analyzed using Interpretive Phenomenological Analysis, allowing for the application of double hermeneutic in which the researchers derived meaning from the lived experiences of participants. **Results:** The analysis yielded five themes in total. Three themes were related to gaps in healthcare systems: Support Needs in Childhood, Support in Transition into Adult Services, and Worry for the Future; and two themes were linked with support needs: Associative Disability in Family Members; and Stigma. All family members had caregiving roles, and these had similarities and differences according to the relationship with the care-receiver. Participants recognized their families were survivors, however maintained a family tragedy rather than positive change outlook. **Conclusions:** Recommendations derived from the findings to alleviate the stressors of the situation for family members include increasing community support and age-related respite facilities. Additionally, improving and enhancing education of disabilities in schools, and immersing and further integrating individuals with disability into society, will alleviate the alienation, isolation and loneliness experienced by family members.

## 1. Introduction

Living with a child with a developmental disorder can be rewarding in terms of supporting capabilities and enabling them to achieve their potential, and it can also be challenging [1,2,3]. Caregivers’ dependency on healthcare systems for support do not diminish during maturation and transition into adulthood [4]. Additionally, despite profound disabilities, offspring sometimes outlive their parents, which continues the impact and responsibilities of siblings, as they ensure the intensive and lifelong support needs of their brother or sister are met [5,6]. Helping with activities of daily living, medical care and management of challenging behaviors can pose significant emotional, social and financial demands, and is associated with poor physical and mental health, although this can be buffered by high social support and low financial pressures [7,8,9]. Models of family illness caregiving and studies of the experience of caregiving indicate that it is a stressful situation [2,10,11,12]. The perspective of family caregivers in severe developmental disability has typically been negative, associated with tragedy, chronic sorrow and persistent hopelessness [13,14,15]. More recently, some studies in the developmental disability literature have explored potential for finding meaning and growth in caregiving parents [16,17]. Young et al.’s [17] comprehensive conceptual model of parental caregiving of children with severe developmental disability (mean age 7.55 years) supported earlier findings of rewards in caregiving—such as reciprocated love and pleasure from seeing their severely disabled child achieve new skills. Findings from this Australian Interpretive Phenomenological Analysis (IPA) study [17] nevertheless revealed that the parents’ experiences were driven by problem-focused coping, and ongoing grief and persistent challenges from unmet service needs, particularly when faced with child deterioration and behavioral issues not improving. That is, whilst the study used a salutogenic approach, and the parents reported that they were able to create a sense of meaning regarding their circumstances over time, this never outshone their frustrations and distress.

Quantitative studies have illustrated the complexity of determining predictors of family caregiver distress in developmental disability when there is a substantial range in social and support characteristics as well as personality of both parents and child [11]. Where both parents are included, studies tend to compare outcomes and show that stress and depression are higher, and health-related quality of life is lower, for mothers than fathers [18,19]. Sen and Yurtsever [20] found sensitivities that can add to the impact on mothers insofar as 40 of 103 mothers in their survey study reported that they had been blamed by other family members for having a disabled child—with half of this blame coming from their husband. Davis and Manago [19] proposed that mothers also blame themselves, and explained this condition in terms of shame and moral distress. Their theory is that this originates in mothers accepting more responsibility for the situation because the child’s initial development was in utero, and mothers also typically have more involvement in child-rearing practices.

The literature that explores the experiences of fathers of children with developmental disabilities is relatively sparse, considering that in many parts of the world, fathers are now heavily involved in the upbringing of their children. This may be a consequence of traditional assumptions that mothers are the main caregivers and fathers are the breadwinners. Nevertheless, there are negative consequences for fathers who have a child with a disability [21,22]. Ogourtsova et al. [23] argued in the title of their study that “Fathers matter”, as they found that communications with healthcare professionals were not always sufficiently attentive. This had multiple consequences for the fathers in their study (n = 7), as well as their child with support needs (aged 18 months to 15 years). In a well-designed comparative study, Alareeki et al. [24] interviewed fathers of young children with and without a diagnosis of autism. The findings indicated caregiving distress; however, unlike in similar studies with mothers, there was an absence of blame. Fathers described feeling poorly understood and having limited social networks because their lives revolved around working in employment to provide for family needs and working at home to support their children. It also emerged in this sample of working fathers that there was reluctance to disclose their children’s autism, and that they had feelings of guilt and shame, stigma, from considering that they did not contribute enough to the family wellbeing [24].

Regarding siblings, the literature is also much smaller than mothers, despite the reality that brothers and sisters generally have longer and ultimately influential relationships with their siblings with support needs. Lee et al. [25] recently conducted a ten-year qualitative review and meta-synthesis of the experiences of siblings of individuals with developmental disabilities, which confirmed that there is a need to support the health and wellbeing of siblings. In addition to the nine studies included in this review, there are earlier studies suggesting that quality of life for siblings of children with intellectual disabilities could be improved [1,26,27]. There is also an emerging emphasis on developing interventions, which have been subject to review [28,29]. An early review of the sixteen quantitative studies of the sibling-focused support services in the US for the 6–12-year-old participants found that two-thirds had positive outcomes [28]. Tudor and Lerner [28] suggested that the focus on services for siblings at this time of life was a start, but that it was vital that more research be undertaken to develop assessments and services because there are significant consequential costs from the negative impact of family caregiving on the quality of life of siblings. A subsequent systematic review found 24 intervention studies involving siblings of individuals with neurodevelopmental conditions [29]. Although the age for inclusion into the studies was higher (4 to 29 years), the oldest participant in any of the studies was 22 years old. The review identified ten mental health effects of family caregiving at baseline for siblings, and three positive post-intervention outcomes in self-esteem, social wellbeing and knowledge of neurodevelopmental conditions—although the latter was only relevant for the younger participants. Critically, the authors noted that all the intervention studies were underpowered, and the improvements seen were generally maturational rather than measured with a comparison control group. Ultimately, the need for more qualitative studies to ascertain what support would be most beneficial was indicated.

Whilst that point remains, a challenge for considering support needs of younger siblings is that many studies have broached this by recruiting sibling participants that are still very young, and thus understanding the impact of family disability on such brothers and sisters is limited to a narrow window of childhood. Moreover, data collection typically involved parents. For example, in a sequential explanatory mixed methods study, Burke [30] collected data from parents on behalf of the sibling children, as well as from the siblings. Nevertheless, Burke discerned that 80% of the siblings actively participated in the caregiving process, and 74% of families found it difficult to do pleasurable activities together. Interestingly, the interviews revealed that many siblings perceived themselves as being disabled, because social activities were limited for them compared to young people unaffected by disability. Also, many siblings become subjected to a form of neglect from their parents because of the overwhelming needs of the disabled sibling. Burke proposed that associative disability can arise through being brought up with a disabled sibling, and that it may persist throughout life. Support for this suggestion can be seen in a study of middle-aged people with a sibling with a developmental intellectual disability in which a form of associative disability was also described [31], which suggests that this observation should be examined further.

A gap in the literature is an in-depth qualitative inquiry of how family members living with a disabled relative may be differentially affected by disability, and why. Thus, the aim of this study was to add to the literature in terms of investigating the personal experiences of mothers, fathers, brothers and sisters providing family caregiving for a relative with severe developmental disability living in the family home. To our knowledge, only one other study has included all types of family members in a severe disability caregiving context [32]. The focus of that qualitative study, however, was to determine the quality of healthcare services in Sweden provided to adults with profound intellectual and multiple disabilities, and most were not living in the family home. Interviews were held with seven mothers, two fathers, two brothers and one sister of twelve different care-receivers who all had formal care support packages. Seven lived permanently in residential care, four lived in a private home with a personal assistant, and just two lived with a parent and a personal assistant. Interestingly, the authors reported that although healthcare support for their profoundly disabled children was available, and notionally there was a shift towards healthcare professionals working in partnership with parents, family members were typically not seen as partners, siblings, they had very little contact with their profoundly disabled family member, and that family involvement in the care of their family member was complicated by healthcare systems who also did not communicate with each other. Recommendations were better recognition of the need for holistic care, and that parents have specialist knowledge that should be acknowledged.

The focus of this study was informal family caregiving where all the care-receivers lived at home, which is the norm in the UK, and healthcare services seek to support the severe developmental needs of the family member. Objectives were to include some participants from the same household to ascertain whether viewpoints accord across parent and sibling, and to gain an understanding of the experience of family caregiving for siblings throughout their childhood into adulthood and its impact on their futures. Thus, an effort was made to recruit siblings over the age of 18, still living at home, for this purpose. The context of associative disability raised in the literature [30,31] also warranted further exploration, and a further objective was to investigate this phenomenon. To account for positive and negative fluctuations in motivations and experiences of family caregivers over time, we discerned that the best approach was to use Interpretive Phenomenological Analysis (IPA) [33]. IPA is a methodology frequently used in health psychology [34]. It draws on phenomenological analysis and hermeneutics to provide an explanation of how experiences support an individual’s sense-making over time [35,36]. This approach has been successfully used in several caregiving studies [17,33,37,38]. In addition to the previously described IPA study of parental caregiving of young children with severe developmental disability, there is a study including family caregivers supporting adults with intellectual disabilities through bereavement processes (n = 7) [37]; and a study of older fathers’ experiences of differentially giving care to an adult child with developmental disability at home in the context of having at least one other non-disabled adult child (n = 5) [38]. Hence this approach provides an established methodology [33,34] to incorporate, for the first time, in-depth interviews to gain an understanding of past, present and future perspectives of giving care to their family member, including the use of support systems, and perceptions of associative disability.

To summarize, the aim of this study was to provide an in-depth investigation of the personal experiences and perspectives of mothers, fathers, brothers and sisters involved in family caregiving for a relative with severe developmental disability using IPA.

## 2. Methods

### 2.1. Design and Setting

This qualitative study employed IPA as a reliable and valid research methodology for exploring subjective experiences and social cognitions of people giving care to a family member with a developmental disability, and construct an idiographic understanding of their lived experience, and how they interpret it within the social context in which they live [39]. The theoretical foundations of IPA are derived from phenomenology and hermeneutics [35] and include an appreciation that a researcher’s interactions with their participants—in conversation at interview, and the resulting transcript of that conversation—have an interpretative element. That is, there is a double hermeneutic.

### 2.2. Participants and Sampling

After obtaining ethical approval, recruitment was undertaken in UK National Health community groups for people with severe disabilities and their families living in Northern Ireland. A formal healthcare prescription is required to be able to attend these groups, which provides acknowledgement of the family caregiving situation, and the availability of support they can access.

The purpose was to recruit a culturally homogenous, voluntary sample of two mothers, two fathers, two sisters and two brothers. Information about the study was presented at several group meetings, contact details were provided, and a detailed Participant Information Sheet was emailed to family members who expressed an interest. This stated the inclusion criteria: over 18 years of age, a blood relative, and living with a disabled child or sibling; and the exclusion criterion: having a disability. From those who confirmed an interest in the study, there was targeted recruitment of eight people to represent the different family relationships, including representatives of severe physical and mental disabilities. The size of the sample (n = 8) was guided by Smith et al. [36] and in line with other studies that have used IPA, and, in practice, there was a small pool of potential participants. Relevant information about participants and the developmental disability of their family member is given in Table 1. Mother-2 and Sister-2 are from one family, as are Father-2 and Sister-1; the data collected in these interviews refer to the same family member.

As is normal in the UK, all the families in the study received standardized financial and social healthcare benefits determined by the level of support needed by their disabled family member. This, in practice, is at subsistence level and does not account for loss of income when a family member is not in the workplace because of caregiving demands. Although participants suggested they were managing economically, it was clear that some events that have become normalized in the western world (such as holidays and meals out) were not possible for these families. All the family members interviewed in this study completed their education and had qualifications for the work they were performing (or was performing) and had no diagnosed health conditions.

### 2.3. Data Collection

A semi-structured interview guide (developed by the authors) was used in a flexible way to explore the experiences of the participants and to allow probing for additional information, when relevant, to search for changes in support needs over time from the care-receiver’s maturation and the caregiver’s own coping, growth or interaction with support provision. Thus, the interview guide contained open-ended, non-directed questions in five areas:Information about the severely disabled family member—their personality, capabilities, independence, activities, caregiving needs and sources of support;Type and extent of caregiving support given by self and others to the severely disabled family member, and feelings about this: past, present and future;Relationships: family, friends and relevant others;Family life, school/work and associated ambitions;Impact on self and other family members of supporting care-receivers.

For participant convenience and choice, consent forms were sent and returned by email, and the interviews were conducted online, via webcam using an online video conferencing platform, at a mutually convenient time when alone and able to talk in private. All the interviews were conducted by the first author. They lasted between 37 and 72 min and were recorded. At the close of the discussion, participants were given the opportunity to reflect on their responses in the interview conversation and qualify or add information.

### 2.4. Data Analysis

Transcription of each interview recording followed the Transcription Standards of Mergenthaler and Stinson [40]. Each participant’s account of their life in the family caregiving context was then subjected to IPA [36], ensuring the subsequent interpretation targeted the lived experience of each of the eight family caregivers [41]. The objective was to develop a comprehensive account of themes that were meaningful in the original transcripts. Thus, bottom-up connections could be made from each semi-structured interview discussion, before being interpreted with knowledge of the caregiving literature and its gaps.

The recorded interviews were transcribed both electronically by the video conferencing software, and independently using the audio recording without the transcript, to pick up on any poor transcription, and to become immersed in the discussion. With accuracy of the transcription assured, each transcript was read twice then analyzed independently to identify relevant items, code into meaningful segments, and classify into emerging themes. The use of clear, annotated coding, which included reference to the available body language and interviewer reflections immediately post-interview, allowed distinctive personal phenomena to emerge. At all stages of the coding process, several rounds of reflection and reference to the transcripts were included to ensure the final themes accurately represented the original interview transcripts and enabled a case-by-case understanding of divergence as well as convergence in the four types of relationship to the care-receiver [41]. Trustworthiness was sought through ensuring suitable recruitment to the study, proper researcher–participant relationships, rigor in transcription and analysis, and a reflective journal was kept by the first author throughout the study. The analysis was checked by the second author, and regular meetings were arranged to critically discuss the progress of the transcription, coding and theme development. Quotes were chosen to illuminate the contribution to the interpreted theme—not simply for prevalence.

### 2.5. Ethical Considerations

The research was conducted following the principles of the Declaration of Helsinki to ensure individuals’ right to provide informed consent, and the prioritization of participant welfare over research interests. In addition, the researchers obtained Institutional Research Board approval (LHUPSY18001680-1). The study aims and associated information were reiterated to participants before data collection and, in addition to receipt of the digitally signed consent, verbal consent was recorded at the time of data collection. The interviewer assured participants that their data would be kept anonymous, and they agreed to the data-identifiers used in this article. Participants rights to withdraw from the study at any time, to ask for a break at any time, and to pass over questions if too challenging were clarified at the start of data collection, in addition to the written information about the study provided.

## 3. Results

Analysis of the interview data revealed that family caregiving in the context of severe developmental disability is initiated and continued because of love and perceived duty. In this sample of mothers, fathers, sisters and brothers giving care to a severely disabled family member at home (see Table 1), there was a strong ethic of ‘family first’ throughout, regardless of family relationship. There was also a clear acceptance of the context in which they lived, alongside understanding that family caregiving inhibits conventional family life and personal opportunities regardless of whether a parent or a sibling. Support needs that would improve quality of life were indicated in all accounts. Altogether, the analysis of the data yielded five common themes: three that were directly associated with gaps in healthcare systems for enabling their caregiving—Support Needs in Childhood, Support in Transition into Adult Services, and Worry for the Future—and two themes that emerged as consequences of insufficient support—Associative Disability and Stigma. Each theme is elaborated upon below, with similarities and differences according to role included in the analysis. Interpretive assertions are supported by illustrative quotes, and reference to the literature where appropriate.

### 3.1. Support Needs in Childhood

Gaps in formal healthcare services were evident in terms of support for family members right across the lifespan. Participants reported challenges for themselves from the time of birth and into unknowns regarding the future. A particular gulf observed was the decrease in support for disabled individuals after school age.

All four parents reported a lack of support at the birth or diagnosis of their child and throughout the course of parenting. There were various points of criticism surrounding the helplessness felt, and their need to gather some hope for the future for their new child and the family. Father-2 explained that despite the shock and devastation experienced when their son was born, “*We just felt as though we were completely on our own…The care system is at its weakest at that point*”. (Father-2) This view was shared by Father-1 and Mother-2. Both asked for help and were frustrated by misunderstandings of the support required. This is a critical challenge for coping effectively with what is ultimately a life-changing event. It is well documented that there is enormous grief and stress for parents when they realize that their new baby is severely disabled [42,43]. Similarly, studies that have examined the communications between healthcare professionals and parents looking for a diagnosis and prognosis consistently show that more than half are dissatisfied, despite the publication of guidelines to support healthcare professionals with this task [44]. This study endorsed the findings of Graungaard and Skov [44] in a different culture and emphasized that improvement in the difficult task of effective communications at the start of the caregiving journey can lead to great improvement in coping skills and in quality of family life.

Parents in this study also revealed that inadequate information from healthcare systems to enable them to adequately and knowledgably care for their disabled child had placed a strain on their marriage. Critically, the demands of caregiving had changed the dynamic of their relationship and their previous life expectations. The recruitment criteria for this study were biased to parents whose marriages had survived their challenges; nevertheless, three of the four parents openly announced that this was not the case for many families with disabled children. As Father-2 asserted, “*I hardly know any parents of special needs kids that are still married. A lot of the marriages or relationships just disintegrate. In fact, all my son’s friends, bar one, have got parents that are not together*.”

Three parents who had other young children at this time also expressed concerns over the wellbeing of their non-disabled children. There is no formal support available for healthy children, despite their dependencies. 


*I think how siblings are catered for through it all is fairly weak… I don’t think that there are very good provisions at all. And I imagine, many siblings of special needs children are in need of very serious attention, many, many of them.*
(Father-2)

### 3.2. Support in Transition into Adult Services

Until the age of 19 in Northern Ireland, there is access to age-appropriate respite centers and, whilst in school, children receive many beneficial and necessary therapies. However, schooling can continue up to the age of 18 years, and at the age of 19, anyone who has any continuing need for healthcare is transferred into Adult Services. This was a particularly raw and emotional point for all family members.


*The respite is like a nursing home scenario. My daughter, who is 23, could be in the room beside a 90-year-old. That to me was every bit as hard as realising when she was a baby that there was something wrong with my daughter.*
(Mother-1)

Most parents were aware of the limited facilities for young adults in enabling them to have any meaningful activities, revealing angst about their child’s future quality of life.


*You would just love some sort of bright, happy, playful place that isn’t all about old people and just sitting watching television or lying in bed. You want breakout areas, you want computer suites and interactive play … but unfortunately there is nothing like that here at the moment.*
(Mother-1)

This is possibly because it was an ongoing problem, and thus fresher than earlier issues with receiving support at the start of the caregiving journey for these parents. Besides angst regarding the type of support available not meeting the needs of the care-receiver, the change impacted on home life because of changes in the care-receivers’ education and routine affected some of the care-receiver’s behaviors, including at home. This was also picked up on by the siblings, particularly that the hours dealing with caregiving needs in the family home had increased. There were some scathing reflections associated with misunderstanding the needs and situation for all in the family, as well as the appropriateness of care. Their lack of preparation for the change in support when a child becomes an adult, when still vulnerable and dependent, must be recognized as a support need that is poorly understood. Kanthasamy et al.’s review of twelve studies that had investigated family caregivers’ adaptation to the transition of their dependent with intellectual disability to adult services similarly found very little positivity about the experience for anyone in the family [45]. Critically, that there is a problem is understood; intervention studies should be undertaken to test supportive approaches before and during transition to minimize the current distress for all family members, in addition to providing more age-specific activities in respite care situations.

### 3.3. Worry for the Future

All eight participants voiced significant worry about the future of their family member, if in different ways. Parents’ concerns existed on two levels: fear regarding what will happen to their child when they could no longer take care of them, and the potential for burden on their non-disabled children when they would have to pick up more caregiving responsibilities. Father-1 stated that his family had moved into the city in preparation for the future. The logic was to ensure his daughter has accessible support when he and his wife were no longer fit enough to care for her, and to ensure that her siblings would not be burdened with the responsibility of caring for her.


*We often ask ourselves—would it be better if we died first or if she died first? It’s maybe a scary thought but I’m not afraid to say it… In all honesty I think I would prefer my daughter to die before us.*
(Father-1)

Similarly, Father-2 revealed that he was constantly considering his son’s future, adamant that he does not want his non-disabled children to be burdened by care requirements. He asserted that this concern was private between himself and his wife, and that his other children do not worry about the future, rather, they simply view their brother how he is at present.

Contrary to the view held by her father (Father-2), Sister-1 revealed awareness that her parents constantly worry about dying and the repercussions for her brother’s life when this happens. She explained that as the eldest child in her family, it was inevitable that she would have the responsibility of caring for her brother after her parents died, and consequently, she has made no plan for things for her future. Sister-2 was also the eldest sibling in their family, and she similarly assumed this responsibility: “*I know what my life will be like. Because I’m the oldest so it will be me… But I’m glad it’s me. I’m able to do it because some siblings will not be able to deal with that.*” Despite their awareness of this responsibility, it was not viewed as a tragedy. Rather, she stated that whatever she did in the future, she will happily include her sibling.

Generally, siblings worried about the future for all their family. Brother-1 said that he constantly worried about what the future holds for his family, and that he had always known that he would become responsible for deciding his brother’s future. However, due to his brother’s challenging behavior, his options are limited, causing him moral distress. He said, “*It is a burden that is essentially going to fall to me and my brothers…It is up to us what happens to him… It is truly heart breaking to think of my brother locked up in a mental hospital.*” Brother-2 similarly related to his unease at putting his brother into a residential care home. However, despite this, he acknowledged that if any family member assumed care for his brother, this person would be picking up a major responsibility. He explained that whilst this is not a new situation for his family, it is not something that a person would expect to take on for their entire life.

Interestingly, there was an unspoken, perhaps unacceptable thought for parents, that their healthy children may also be required to look after them, as well as their disabled sibling at some future point. People with severe disabilities are living longer and increasingly outlive their mother and father [5]. The siblings were aware of this.

### 3.4. Associative Disability

This theme encapsulates the extent to which participants considered themselves to be ‘disabled’ by their association to their child or sibling, for the family and personally. Some form of “*a disabled child is a disabled family*” was uttered by all participants. Regardless of relationship, each was keen to explain their inability to do what they regarded as ‘normal’ family activities, such as going on holiday or for a family meal. However, despite the knowledge and awareness that their family life differed to that of a family not affected by disability, it was clear that there was a lack of certainty on what they were missing out on. For example, Mother-1 noted that “*Her disability has held us back, but how it has impacted us, I don’t know because it is just our normal. It is just the way our life is and always has been.*” Similarly, Brother-1 considered “*I don’t know what other families do to know what I’m missing out on. Like to what extent do you go for drinks with your family, or go on holiday, or for meals? I don’t even know what I’m missing out on*”.

The four parents were all aware that their child’s disability could be impacting the lives of their non-disabled children, and that they had missed out on many opportunities. Mother-2 spoke of her constant guilt that her two other children could not be a part of normal family activities. She believed life was a lot more stressful for herself and her family than life in a family unaffected by disability. She spoke at length about the many responsibilities of her non-disabled daughters that other children of their age did not have to face. Similarly, Father-1 told us that “*We nearly killed ourselves to make sure they didn’t miss out*… *But the fact is that they have missed out on having a carefree childhood because they had a little sister that was disabled*.” And Sister-1 articulated, “*My brother is the priority. He will always be the one that comes first. I think having that dynamic in a family is quite weird*.”

All siblings said that their parents had done everything in their power to afford them the opportunities to fulfill any personal aspirations, but alongside this, they also stated that they were not granted the same liberties as their peers unaffected by disability. Siblings outlined limitations on their social life and hobbies and stated that they often had to cancel plans due to their caregiving responsibilities. They saw little benefit from the tasks and limitations of being in a caregiving situation. Brother-1 specified that due to his brother’s challenging behavior, there was always a need for one of the three non-disabled brothers to be at home as his parents could not manage an aggressive episode alone. “*Something we 100% follow is that all three of us cannot be out at the same time. That is absolute. One of us needs to be there at all times*.”

All four parents considered that their life was different than the lives of parents of non-disabled children and did not readily take any credit for the benefit added to their disabled offspring’s life. Mother-1 was particularly negative in her tragic recollection of an event that made her realize that the children of her siblings and friends were now grown up. Those parents were out enjoying life because they now do not have to think about who is looking after their children, whereas she would never be afforded this luxury and her life will increase in difficulty as she ages.

Another sign of associative disability was early maturation of the siblings. Sister-2 explained that her childhood was quite traumatic. “*We weren’t babied at all, despite still being young, because my parents had bigger things to worry about, like: ‘Come on! There is someone else who needs more attention’*”. (Sister-2)

Parents also acknowledged their other children had to grow up quickly and have responsibilities and stresses in their life at a young age.


*My non-disabled daughter had to grow up a lot quicker than most other children and she did take an awful lot on when her sister was young… She definitely matured a lot younger than most children… As a parent, I don’t think I give her enough credit for how much she does and how much she has to do compared to children in other families.*
(Mother-1)

Whilst there has been some concern about mental health impacts of early caregiving, the outcomes have not shown this was necessarily so [46,47], and overall, this was not evident in this study. Nevertheless, it remains that siblings’ quality of life could be improved as they did not take pleasure in their responsibilities. There are educational and social consequences and time-dependent life chances that become unavailable to them. This was recognized by all. Both siblings and parents talked about the loss of a professional career due to caregiving requirements. Parents explained that an inability to find suitable childcare for their disabled children meant their careers took a ‘backwards step’. Brother-1 spoke about the loss of his parents’ careers, as both had taken early retirement to care for his brother. “*I would say my dad’s life is just 14 hours a day looking after my brother… That’s all I could say for his life*”. Father-1 reported a similar situation for his wife: “*her career has suffered, and I do know she would mourn the loss of her career. For all she loves our daughter, her career took a massive turn. It was like a fork in the road and that’s impacted her*.”

All family members recognized an unnatural hierarchy of attention due to prioritizing the care-receiving member, and there seems to have been a resignation to it. That is, despite stoic acceptance of the way family life was, and evidence of adaptive coping, they refrained from giving any positive examples of their resilience and preferred to portray their family life as tough. Although it has been suggested that, over time, parents go through a process of coping and finding positive meaning in raising a disabled child [17], there can be a critical window for some social and career-related opportunities, and for these participants, realizing their lost opportunities were never forgotten.

### 3.5. Stigma

All participants wanted to discuss the need to tackle ongoing societal stigmatization of disabled people, and the repercussions it has on all family members. Although they did not recognize themselves as stigmatized, they were affected by overt negative behaviors directed at their family member, which impacted on their own identity. They were conscious of people staring at, laughing at, and whispering about their visibly severely disabled family member, and explained that this could be extremely frustrating and an upsetting experience for them. Brother-1 became quite agitated when he explained a situation during which his brother had come to watch him play in a football match, and other players pointed and laughed at him.


*I can see people judging him and it really annoys me even though I know that he can’t see it. But I can see it! He is so oblivious to it, but I am not, and I am stood there watching them laughing at him.*
(Brother-1)

These experiences alienated him, and evoked extreme upset, anger and the overwhelming urge to confront or lash out at those involved.


*Why stare? Why do that? Why stare right at the person? It just alienates both the person and the person they are with …You really just want to go over and hit the person laughing. You just get so angry.*
(Brother-1)

Most participants could describe a similar negative emotional response when faced with this type of reaction, and their experience was that disabled people are often the target of many inappropriate and insensitive jokes. All participants stated that they were not ashamed or embarrassed of their family member, but that they did not openly speak about them or their struggles. The siblings explained that they would rarely disclose information about their disabled sibling and tend to keep their peers on a ‘need to know basis’. “*I don’t use friends to talk about things that are really sad or annoying in terms of my brother, I would just go to my parents for that*.” (Sister-1). Sister-2 commented that whilst wider knowledge of her home life could have made her life easier, it would have hindered her progression or acceptance for her own merits. Specifically, she was clear that she did not want to be treated differently or be known as the girl with the autistic sister, “*I am not the one with special needs, I do not need to be treated as though I have special needs*”. Similarly, Brother-1 revealed that he would not speak about his brother as he did not want to be seen to be looking for sympathy. He likened his situation to people not speaking out about having poor mental health. “*It’s just my burden to carry, it’s my family’s issue to deal with so I don’t really chat much about it.*”

This approach was perhaps a reflection of their parents’ stated views that people do not want to hear bad news. Certainly, the four parents were consciously not oversharing their struggles. Father-1 recalled a conversation between himself and his boss at work. She had complimented him on not being a typical parent of a disabled child who needed to come to work and unload his issues. She liked that he was simply there to work and did not bring his home life or daughter’s disability into the workplace. Consequently, he said he wouldn’t share every mini crisis and was very conscious of that balance.

A corollary of this coping approach, however, was that participants said that they constantly experienced a sense of isolation and loneliness. It could be a challenge for introducing more talking support opportunities if there is an observed general reluctance to talk about their home life and thinking that the only people who truly cared about and understood their plight were those living in similar situations. Sister-2 stated that she often felt detached from and unable to relate to her peers as they concerned themselves with trivial issues and did not understand the greater challenges she faced. Again, this may have come from parents’ own views, as expressed by Father-2: “*It is only really other people who have got extreme situations at home who really understand. I rarely talk about it, because most people don’t want to understand*.”

## 4. Discussion

This original study investigated the phenomenon of caregiving for a family member with severe developmental disability for parents and adult siblings. The aim of the study was to investigate the personal experiences and perspective of mothers, fathers, brothers and sisters using IPA, towards providing recommendations for intervention for these family caregivers to relieve their distress and exhaustion. Our findings replicate and extend earlier literature with this care-recipient group in a new context—considering whole family caregiving from the perspectives of mothers, fathers, sisters and brothers giving care to a severely disabled family member over time—and they provide new insights for the caregiving literature in terms of associated disability and stigma impacting on behaviors and quality of life of all family members because of living with a disabled family member.

The use of IPA as the study methodology was purposeful. Mindful of recent criticism in the disability caregiving literature of research that solely focuses on what has been termed a tragedy model, and not sufficiently delving into opportunities for personal growth over time across the family [17,48], IPA allows for positive and negative aspects of caregiving to be revealed, and it can account for fluctuations in motivations and experiences over time [33], as resilience emerged in this survivor sample. We raise two important points related to this: first, studies that have found enrichment in caregiving contexts [17,48] have also reported that for those caregivers, stress is their major concern [49], and also, that whilst cognitive behavioral therapies may be able to support meaning-making and growth from the caregiving, this does not resolve the distress and grief associated with caring for a severely disabled child [17].

In this research, in which there were clearly high demands to adequately care for a family member with severe disability in the family home [50], accounts were largely negative. Family members recognized that they had developed relevant resilience skills, and they had obtained the resources to be able to survive, but these did not promote thriving in the context they were living in. There was a sense of success that was related to staying together as a family unit, and what we interpret as a growth in resilience in the context of new challenges that emerged over time [51]. Nevertheless, it would be difficult to endorse Hastings’ [48] assertion that the common negative narrative around the experience of living with a child with developmental disabilities should be rejected. In line with many other studies, we found that caregiving was driven by love and duty [11,52], and there are recommendations we can make from the support needs identified from the IPA. Positives that were raised were in relation to early maturation in siblings. In line with previous reports from parental observations [53], and parents in this study, both sisters and brothers reported an intrinsic duty of care over their sibling, which resulted in early maturation. Whilst this concept can be viewed favorably [51,54], and the four siblings in this study considered it to have some benefits, the fact that it arose from chronic stress and exposure to serious responsibilities at a young age cannot be thought of as positive. It remains that siblings of children with severe developmental disabilities should be of concern to healthcare services as their needs can be hidden [54]. Siblings are subject to significant social exclusion, responsibility and emotional hardships, all of which can lead to poor psychological health. If left unsupported, siblings can develop poor self-esteem, have limited social interactions and a poor quality of life. However, through implementing effective support strategies, the general psychological wellbeing of siblings can be improved [27].

Burke [30] suggested that living with a disabled sibling generates a disabled identity for brothers and sisters, and that many siblings become subjected to a form of neglect due to the overwhelming needs of their disabled sibling. Their assertions were supported in this study. There were reports of lower levels of social interactions and an impaired social life compared to other young people unaffected by disability. Also, in an extension of Burke’s theory, we found that all family members had challenges to do perceived normal activities; logistical and time demands from caregiving meant they missed out on opportunities generally available to other families. This was recognized as associative disability in action, and common to all family members in this study. In siblings, it was not limited to a specific period in their childhood, but it extended into adulthood. Sibling participants considered their caregiving responsibility would continue into old age, as they would take over from their parents to maintain family caregiving—which is what has been reported elsewhere [31].

All family members confirmed that they did not speak openly about their disabled family member; however, their reasoning for this differed. Siblings did not wish to be treated differently to their peers because they had a disabled sibling, whereas parents did not disclose information about their family life as they perceived that friends and work colleagues do not appreciate complaining or want to hear bad news. Ultimately, all participants disclosed a consequent sense of isolation that arose because their peers did not understand their plight. It has been argued that disabled people are among the most socially excluded groups and endure discrimination and prejudice for the duration of their lives [55]. Evidence from this study allows us to declare that this social exclusion extends to family members who relate to being disabled by association.

All participants reported negative experiences directed at their disabled family member when out in public. The evident stigmatization was reported as upsetting and frustrating and it served to marginalize and alienate them from society. Stigma is a long-standing societal issue. It has been argued that to stigmatize is a natural human reaction to the unknown, and often the perpetrator stigmatizes as an act of self-preservation or psychological defense [56]. Yet such intense and engrained stigmatization within society when harm is unlikely is detrimental to the psychological health of community members and leads to social exclusion and segregation [57]. Our findings indicate that the negative consequences of social stigma can be seen in family caregivers, as alluded to in Nurullah’s narrative study of parents of children with developmental disabilities [2]. Nevertheless, there are studies which suggest that such stigma is not inevitable. For instance, Alshaigi et al. [58] found only one-third of parents of children with autism experienced stigma, and Bala et al. [59] reported relatively low levels of stigma in siblings of children with intellectual disabilities compared to parents. These studies suggest that interventions are possible where it arises. There have been high school interventions to increase knowledge and improve attitudes regarding disability to reduce the stigmatization of disabled people [60,61]; however, these are not included in mainstream schooling in the UK, where this study took place. Recommendations from these international studies [60,61] should be considered for action everywhere.

Formal support for participants in the current study was perceived to be totally insufficient. In line with other literature [32,42,43,44], parents were very disparaging about communications with them from healthcare professionals, and particularly the absence of signposting to receive the information and support they needed after the birth or the diagnosis of their child. The literature indicates that this is an international problem. Recommendations to improve this issue [44] have yet to filter through sufficiently to set caregiving on a stronger footing. Similarly, the complex needs of many disabled children meant that the families interacted with several types of support workers, but the fleeting visits with many different professionals specializing in different areas results in uncertainty regarding who to contact when specific issues arise [62].

The move from education and child support services to adult services when the family member reaches the age of 19 was substantial for these participants in this situation. The critical issue is that adult services provide considerably less interaction, activity and education than when in the school system, and that adults services are mainly focused on much older adults and respite care. The transitional stage, in particular, was experienced as stressful by the care-receivers, and caused an increase in family caregiving. Although Cowen et al. [63] proposed an approach to funding personalized transition to adulthood for young people with complex needs, their proposed framework has not been adopted across the UK, and studies that have cited this work have focused on the outcome for the disabled adult in isolation. A similar approach, tested in a PhD thesis in the US [64], also has not been rolled out as standard, to our knowledge. For parent caregivers, there were feelings of abandonment regarding support for their family member with disability after school age, as well as themselves.

Both parents and siblings expressed anxiety regarding their family’s future and the magnitude of the responsibility they possess. Adding to previous findings with a sample of mothers [65], we found that fathers, as well as mothers, expressed concern for their disabled child’s future. They reported constantly worrying about the quality of life that their child will have when they are not able to care for them, combined with the desire that their non-disabled children are not burdened by caregiving responsibilities. Although all parents have a duty of care for their children, for parents of children with developmental disabilities, there is no stage of reaching independence. Their responsibility is heightened, and it remains for the duration of their life or their child’s life.

Sisters and brothers also worried regarding their sibling’s future, but in a different way to their parents. Their worry was aligned to a sense of inevitability that they will assume responsibility for their sibling when their parents are no longer fit to provide care. For them, planning for their own futures was difficult. Whilst not unconcerned about their disabled sibling’s welfare, it was not their focus. Both sisters and brothers reported inner turmoil as they acknowledged that they would accept caregiving responsibilities, whilst simultaneously coming to terms with the gravity of this requirement and the consequences for their life. This finding supports assertions that siblings experience stress, guilt, and confusion when they realize that they should be involved in their sibling’s future, whilst simultaneously lacking the knowledge and certainty of what awaits them and attempting to fulfill their expectations and desires for their own life [66].

Ultimately, many recommendations can be made in attempt to improve support to diminish the challenges faced by families affected by disability and improve their overall quality of life. Some of these reiterate recommendations already in the literature for some years, but are unfulfilled. This could be because caregivers are typically hidden patients [11,67]. Based on this study and the literature, we must recommend that the support available to families supporting severe disability is re-evaluated, that suitable and sufficient support and guidance is made available, and that families are explicitly informed about the services available to them [62]. We found that confidence in support healthcare services was not high because of poor experiences at the start of the caregiving journey, and stigma impacted caregivers such that they were reluctant to disclose all their needs. Development of interventions to provide better information and support at diagnosis, and resources to put recommendations into place, are strongly recommended. It is also recommended that the transition from child to adult services is improved and attention to the family situation is incorporated. In particular, healthcare facilities for young adults should be enhanced to ensure they are appropriate for them, which in turn will reduce angst for the future and increase respite opportunities to prevent associative disability [29]. It is recommended that support groups for young caregivers are formed in schools to provide siblings with the opportunity to speak to others who can relate to their situation. This will reduce feelings of isolation and improve overall psychological wellbeing [27,45].

It must be recognized that there are limitations to this study. Whilst a lack of diversity is typical of IPA studies where small homogenous samples are used to understand a phenomenon and reduce confounding cultural explanations, it remains like that because all participants were recruited from the same background (Caucasian Northern Irish Catholics), and the situation may differ for some other contexts. Also, the study recruited a survivor sample, who were willing to explain their experiences in an anonymous research setting. Many parents of children with developmental disabilities become separated or divorced [68]. This implies that the study may not have identified the full extent of the support needs related to successfully coping with raising a disabled child or sibling, and many people may face even greater challenges than this study has been able to identify. Data collection undertaken with the help of video conferencing has strengths and limitations [69]. In this study, it supported recruitment, as participants appreciated being able to talk in their own home, with minimal additional time costs.

Future studies could use mixed methods to broaden the scope of the study—particularly regarding an exploration of triggers for breakdown of family caregiving and break up of families who are caregiving in the context of severe developmental disability. We recognize these are significant gaps that remain in the literature regarding suitable and sufficient healthcare resources for families with a severely disabled child and other children through the lifespan. We also suggest that future studies should develop and evaluate intervention programs on themes identified in this study to support family caregiving in the context of severe developmental disability.

## 5. Conclusions

Family members of an individual with a developmental disability experience significant psychological distress. Although we were able to discern some positives from the family caregivers’ accounts, these could not outshine any of the articulated caregiving distress. There were convergence and divergence in caregiving experiences according to family relationship; nevertheless, striking commonalities were seen insofar as gaps in healthcare support, associated disability and stigma had a comprehensive impact on the quality of life of all family members. The five themes identified can readily lead to developing improvements to support for such family caregivers. These include improving timely communications to parents at diagnosis and improving services for care-receivers when they become young adults. There is a need to promote a change in attitudes and community support to alleviate the challenges of associative disability and stigma for family caregivers to enable them to thrive. We also recommend introducing quality education of disabilities in schools to alleviate the stigmatization of disabled people, and diminish the alienation and isolation experienced by family members.

## Figures and Tables

**Table 1 healthcare-13-02550-t001:** Participant and care-receiver information.

Participant	Care-Receiver Diagnosis	Nature of Care-Receiver Disability	Care-Receiver Age	Typical Caring Time
Mother-1	Cerebral Palsy and Learning Disabilities	Very poor balance and speech. Communication level of a 5-year-old.	23	4 h/day
Mother-2	Autism Spectrum Disorder Level 3.	Limited speech. Emotional and behavioral problems.	9	24/7
Father-1	Cerebral Palsy	Wheelchair user. No speech. Mentally capable. No fine or gross motor skills.	18	24/7
Father-2	Mosaic Tetrasomy 8p	Occasional wheelchair use. Physical limitations.	11	1 h/day
Sister-1	Mosaic Tetrasomy 8p	Occasional wheelchair use. Physical limitations.	11	1.5 h/day
Sister-2	Autism Spectrum Disorder Level 3.	Limited speech. Emotional and behavioral problems.	9	2 h/day
Brother-1	Cerebral Palsy	Wheelchair user. No speech. No fine or gross motor skills.	29	30 min/day
Brother-2	Autism Spectrum Disorder Level 3. Obsessive–Compulsive Disorder	Mental capacity of a 5-year-old. Emotional and behavioral problems.	23	3 h/day

## Data Availability

The raw data supporting the conclusions of this article will be made available by the authors on request.

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
