# Peer review of "Understanding the Support Needs of Family Caregivers Living with Severe Developmental Disability: An Interpretive Phenomenological Analysis"

_healthcare, 2025, doi:10.3390/healthcare13202550_

Round 1

Reviewer 1 Report

Comments and Suggestions for Authors

This is a very well written article that lays out some of the key concerns of family members providing care and support to persons with severe disabilities. I particularly like the emphasis on siblings, because I believe they are often an overlooked group.

My only criticism of the paper, is that I don't see things that are new here. Many of the issues are explained in the references cited -- and there are more -- including whole books on this subject. For example, by one of the people they cite, Burke, (2003) Brothers and Sisters of Disabled Children, and Connors and Stalker (2010) The Views and Experiences of Disabled Children and their Siblings.

THere are also a whole range of books devoted to supporting siblings (Jessica Kingsley Press has many) -- and there are organizations like the Sibling Support Project (https://siblingsupport.org/) that also have lots of materials, workshops, and resources -- and can give access to many, many sibs.  If one looks on their social media you can find a broader range of responses than those of the people mentioned in this study -- in terms of their relationship to their family member, their attitudes towards care and support and their non-disabled family members, and the issues they face. That is not a random sample by any means but it does make one wonder a bit about selection bias in terms of which family members were willing to be part of this study.

So I am of two minds here. On the one hand, this article would be very useful to people who are new to the subject and I would like people to have access to it. On the other hand, I don't think it advances the knowledge of people who are working in this area.

This comes down to the editors of the journal in terms of the purpose of its publications. I will say "accept" only because the article is well written and referenced and I'd be fine with it being out in the world as is.

Reviewer 2 Report

Comments and Suggestions for Authors

I want to begin by expressing my heartfelt thanks for the opportunity to read and critique your manuscript—truly, it’s an honor. Your research topic is wonderfully timely, and it dovetails nicely with current academic discussions. I hope the comments below help refine your core arguments and elevate the overall impact of your paper.

ABSTRACT

Your background section does a great job of highlighting the broader importance of the study and setting the geographical stage, but it might feel even more grounded if you briefly mention why Northern Ireland is such a fitting location. Are there specific local challenges, for instance, that might not be as common elsewhere? In addition, the objectives are outlined clearly, yet clarifying exactly how identifying support needs will affect real-world settings could give this section an extra spark. 

Regarding methodology, you’ve noted the use of in-depth semi-structured interviews and Interpretive Phenomenological Analysis (IPA). To deepen readers’ understanding, you might share a line or two about why these particular methods best capture family caregivers’ experiences. If you also describe why a sample of eight was sufficient—perhaps touching on qualitative study norms for IPA—readers will appreciate the thoughtful design. 

The results mostly list themes but could be strengthened by adding a brief explanation for each one—how was it identified, and why does it matter? A little extra context can help show how these findings build on existing literature. Finally, your conclusions are already robust, offering practical recommendations drawn directly from the study. You may want to highlight their potential impact on policy or practice. Also, discussing any limitations and hinting at future research directions will provide a fuller picture for anyone eager to take these insights further. 

INTRODUCTION

Consider explicitly showing how you’ll apply interpretive phenomenological analysis to capture and interpret caregivers’ experiences—this helps bridge your literature review and methods. Although you mention the study’s aim, placing it earlier might guide readers right from the start. Stating your research questions or hypotheses up front could also sharpen the focus. 

If possible, reorganize the introduction so it travels from broad context (e.g., general background) to specific issues (e.g., family caregiving challenges), and then lands on the gap in knowledge your study will fill. You do identify a research gap, but emphasizing what fresh insights this project offers could highlight its value. Finally, consider weaving in relevant theoretical models or frameworks, as that can anchor the research in a conceptual base you can refer back to later in your analysis. 

METHODS

In qualitative work, smaller sample sizes often make sense, especially for IPA, but adding a note on why eight participants were just right can reassure readers of your rigor. In addition, if there’s room for future expansion, you might consider increasing participant diversity—varying family roles or cultural backgrounds might yield richer data. 

Thinking ahead, a longitudinal component could uncover how caregivers’ needs change over time, shifting our understanding of lived experiences in a dynamic way. Triangulating methods—maybe diaries or observational data—could also reinforce your interpretations by offering multiple perspectives on the same phenomenon. 

DISCUSSION

Breaking the discussion into subheadings like “Emotional and Social Challenges,” “Stigma and Social Exclusion,” “Support Needs,” and “Recommendations” is a great idea to help readers navigate seamlessly. When addressing siblings’ unique experiences, consider emphasizing their psychological hurdles in a bit more detail—if you have data at hand, even brief narratives or quotes might illustrate your points vividly. 

Don’t hesitate to be explicit about limitations here too: acknowledging what the study can’t address paves the way for future researchers to build on your work. Finally, ensure that each recommendation clearly grows out of a specific finding. By linking suggestion to result, you’ll show exactly how the study’s evidence supports your thoughtful proposals. 

In sum, your manuscript already shows plenty of promise—your passion for the topic shines through, and the study’s purpose is compelling. With a few structural refinements and some added depth, I believe it will make a valuable contribution to ongoing conversations about family caregiving. Thank you for letting me engage in this meaningful work, and I wish you the very best with your revisions!

Reviewer 3 Report

Comments and Suggestions for Authors

Thank you for the opportunity to review the manuscript titled Understanding the Support Needs of Family Caregivers Living with Severe Developmental Disability: An Interpretive Phenomenological Analysis. Overall, this is a well conducted and clearly written study that offers valuable insights into the lived experiences of family caregivers of individuals with severe developmental disabilities. Below is section-by-section feedback.

The introduction is comprehensive. It provides a good overview of prior research across mothers, fathers, and siblings and highlights both aspects of caregiving. It situates the study within gaps in existing research, such as limited attention to fathers and siblings, and the under-explored concept of associative disability. The manuscript could be strengthened by the following revisions:

  • More explicitly state the research questions or aims near the end of the introduction to provide clear focus.
  • Consider clarifying how the study adds to or differs from existing IPA studies, beyond its inclusion of multiple family members.

The methods section is detailed. The justification for Interpretive Phenomenological Analysis is presented, and sampling across family members is purposeful. Ethical considerations are addressed, and trustworthiness strategies such as reflective journaling and cross-checking of coding are used. The manuscript could be strengthened by the following revisions:

  • Provide more detail about participant demographics (e.g., socioeconomic status, education) to contextualize findings.
  • Clarify how saturation was determined, given the small sample size.
  • Explain more explicitly how potential researcher bias was managed beyond reflective journaling.
  • Describe how differences between families with multiple participating members were handled analytically.

The results are organized around themes and supported by participant quotes. The five emergent themes are defined. The inclusion of diverse voices (mothers, fathers, sisters, brothers) is a notable strength showing both convergence and divergence in experiences. The manuscript could be strengthened by the following revisions:

  • Provide a thematic map or diagram to visually represent relationships among themes.
  • Clarify whether frequency of themes across participants was considered important or whether depth of meaning was prioritized.

The discussion is contextualized in existing literature and provides important implications for research, practice, and policy. The authors extend concepts such as associative disability and stigma, and they connect findings to gaps in healthcare systems. The recommendations are practical. The section also acknowledges limitations. The manuscript could be strengthened by the following revisions:

  • Strengthen the articulation of the unique contributions of this study compared to prior research.
  • Provide a clearer balance between describing negatives and highlighting resilience or adaptive coping that emerged.
  • Expand briefly on future research directions beyond IPA (e.g., longitudinal or intervention-based studies).

Round 2

Reviewer 2 Report

Comments and Suggestions for Authors

Many thanks for addressing all my comments.